# Influence of C_60_ Nanofilm on the Expression of Selected Markers of Mesenchymal–Epithelial Transition in Hepatocellular Carcinoma

**DOI:** 10.3390/cancers15235553

**Published:** 2023-11-23

**Authors:** Malwina Sosnowska, Marta Kutwin, Katarzyna Zawadzka, Michał Pruchniewski, Barbara Strojny, Zuzanna Bujalska, Mateusz Wierzbicki, Sławomir Jaworski, Ewa Sawosz

**Affiliations:** Department of Nanobiotechnology, Institute of Biology, Warsaw University of Life Sciences, 02-776 Warsaw, Poland; marta_prasek@sggw.edu.pl (M.K.); katarzyna_zawadzka1@sggw.edu.pl (K.Z.); michal_pruchniewski@sggw.edu.pl (M.P.); barbara_strojny-cieslak@sggw.edu.pl (B.S.); s200104@sggw.edu.pl (Z.B.); mateusz_wierzbicki@sggw.edu.pl (M.W.); slawomir_jaworski@sggw.edu.pl (S.J.); ewa_sawosz-chwalibog@sggw.edu.pl (E.S.)

**Keywords:** autophagy, fullerene, inflammation, invasion, liver cancer, oxidative stress, phenotype transition

## Abstract

**Simple Summary:**

Understanding the relationship between hepatocellular carcinoma recurrence and the process of cellular phenotypic transformation is crucial for developing effective therapy. The main problem is that marginal cancer cells with strong migratory activity remain in the post-resection tumour bed. The degraded extracellular matrix of the liver combined with the presence of cytokines causes the abnormal transduction of signals into the cell and, consequently, cell migration through the vascular basement membrane. We have previously shown that growth factors induce the transformation of epithelial cells to a mesenchymal phenotype. In this study, for the first time, the effect of fullerene surfaces on the invasion of HepG2 and SNU-449 cells with epithelial and mesenchymal phenotypes was compared. It is predicted that fullerene C_60_, as a material that inhibits the secretion of the transforming growth factor, may reduce the invasion of mesenchymal cells and not affect epithelial cells by reducing the expression of extracellular matrix proteases. Our study aimed to determine the effect of the fullerene surface on reducing cell invasion by inducing mesenchymal–epithelial transition. Epithelial cells concentrated at the primary site of the tumours are an easier target for radioembolisation therapy.

**Abstract:**

The epithelial–mesenchymal transition (EMT) is a process in which epithelial cells acquire the ability to actively migrate via a change to the mesenchymal phenotype. This mechanism occurs in an environment rich in cytokines and reactive oxygen species but poor in nutrients. The aim of this study was to demonstrate that the use of a fullerene C_60_ nanofilm can inhibit liver cancer cell invasion by restoring their non-aggressive, epithelial phenotype. We employed epithelial and mesenchymal HepG2 and SNU-449 liver cancer cells and non-cancerous mesenchymal HFF2 cells in this work. We used enzyme-linked immunosorbent assays (ELISAs) to determine the content of glutathione and transforming growth factor (TGF) in cells. We measured the total antioxidant capacity with a commercially available kit. We assessed cell invasion based on changes in morphology, the scratch test and the Boyden chamber invasion. In addition, we measured the effect of C_60_ nanofilm on restoring the epithelial phenotype at the protein level with protein membranes, Western blotting and mass spectrometry. C_60_ nanofilm downregulated TGF and increased glutathione expression in SNU-449 cells. When grown on C_60_ nanofilm, invasive cells showed enhanced intercellular connectivity; reduced three-dimensional invasion; and reduced the expression of key invasion markers, namely MMP-1, MMP-9, TIMP-1, TIMP-2 and TIMP-4. Mass spectrometry showed that among the 96 altered proteins in HepG2 cells grown on C_60_ nanofilm, 41 proteins are involved in EMT and EMT-modulating processes such as autophagy, inflammation and oxidative stress. The C_60_ nanofilm inhibited autophagy, showed antioxidant and anti-inflammatory properties, increased glucose transport and regulated the β-catenin/keratin/Smad4/snail+slug and MMP signalling pathways. In conclusion, the C_60_ nanofilm induces a hybrid mesenchymal–epithelial phenotype and could be used in the prevention of postoperative recurrences.

## 1. Introduction

Epithelial–mesenchymal plasticity (EMP) is the ability of cells to convert between mesenchymal and epithelial phenotypes in response to environmental signals such as autophagy, inflammation, hypoxia and genetic manipulation. In cancer, the presence of EMP is a detrimental process that is required for metastasis, chemo-resistance and immunosuppression [1].

The mesenchymal–epithelial transition (MET) is a reversible biological process in which mesenchymal cells lose their ability to migrate and organise into epithelial sheets. Epithelial cells are polarised and have an asymmetric distribution of cell organelles and strong intercellular connections. In embryogenesis, MET plays a key role during the formation of somites and renal tubules, as well as in the restoration of the epithelial barrier in damaged tissues. In carcinogenesis, the introduction of cells into a stable state of the epithelial phenotype creates the possibility of pharmacologically limiting the development of the disease. When complete elimination of cancer cells is not possible, promoting asymptomatic cell dormancy can extend the lifespan of the patient [1].

The epithelial–mesenchymal transition (EMT) is the first stage of cancer cell invasion: resting cells acquire the ability to dedifferentiate, migrate and metastasise, and they become drug resistant [2]. So-called mesenchymal cells with an elongated shape and anterior–posterior polarisation gain the ability to invade the extracellular matrix (ECM) and neighbouring tissues, enter the blood and lymphatic systems and then colonise lymph nodes and distant organs [3]. Mesenchymal cells are separated from each other by the ECM and do not have a basal lamina separating them from neighbouring cells. EMT is an ordered process during embryonic development, but it is highly dysregulated in cancer [4]. Observations of EMT without the full repertoire of canonical molecular changes, such as the retention of E-cadherin expression, are often described as partial EMT or a hybrid epithelial–mesenchymal (pEMT) phenotype [5].

The invasion of tumour cells through the degraded ECM is enabled by matrix metalloproteinases (MMPs), a family of 28 zinc-dependent endoproteases whose function is to degrade ECM proteins, including collagens, aggrecan, fibronectin, vitronectin and laminin. MMPs exist as inactive zymogens that are activated by reactive oxygen species (ROS) and cytokines. MMP overexpression leads to the degradation of the vascular basement membrane and the excessive migration of tumour cells through the ECM. The tissue inhibitors of metalloproteinases (TIMPs) block the action of several MMPs simultaneously, thus inhibiting proliferation and migration. The balance between MMPs and TIMPs maintains the correct architecture of the ECM and thus prevents invasion. TIMP-1 overexpression is associated with hepatocellular carcinoma (HCC) lung metastases, while TIMP-2 expression is downregulated in patients with HCC [6].

HCC originates from hepatocytes and is the fifth most common malignancy in the world. It develops as a result of liver damage and inflammation as well as redox imbalance. HCC accounts for 85–90% of all cases of primary liver cancer. A factor that increases the risk of this type of cancer is hepatitis B and C [7]. Recent studies have reported that inflammation, accompanied by the production of ROS, causes fibrous changes and the development of cirrhosis of the liver, followed by the formation of a neoplastic tumour [8]. In our previous work, we proved that cytokines such as transforming growth factor-beta 1 (TGF-β1), tumour necrosis factor-alpha (TNF-α) and epidermal growth factor (EGF) induce the EMT of liver cancer cells in vitro [9]. Yazaki et al. [10] showed that TGF-β1 increases the production of ROS by inducing NADPH oxidase (NOX) of the respiratory chain, which causes oxidative stress in cells and enhances EMT. Muthuramalingam et al. [11] reported that TGF activation triggers NOX induction. Thus, one of the main causes of liver degradation is the inhibition of the antioxidant defence system and the excessive production of ROS [12]. It seems that the use of a material with antioxidant and anti-inflammatory properties can inhibit cell invasion by restoring their non-aggressive, epithelial phenotype. The main nanomaterials with antioxidant properties include fullerene, cerium oxide, platinum and silica [13]. 

Fullerene C_60_ is an allotropic form of carbon and the most stable form of fullerene. Fullerenes are unique because they act as both a free radical scavenger and a singlet oxygen generator. Fullerene C_60_ is redox-active and induces superoxide anions and oxidative damage to cells and lipid peroxidation. However, it can be oxidised in a biological environment; in the hydroxyfullerene form, it becomes an effective antioxidant that removes ROS [14]. This is due to the delocalisation of π electrons over carbon atoms, which can easily react with free radicals [13]. The high affinity of fullerenes for the liver after intravenous administration and their low toxicity make them an ideal therapeutic agent for liver diseases [12]. The advantages of using fullerenes in cancer therapy are the possibility of incorporating them into artificial lipid bilayers, their strong properties of photoinductive DNA cleavage and their antioxidant and immunomodulating properties [15]. The advantages of C_60_ over other nanosystems and drugs inhibiting metastases are their homogeneity of structure (always 60 atoms); hydrophobic, reactive cage structure; natural origin–combustion product [16]; photostability [17]; low toxicity (even intracellularly [18]); controlled endocytosis of particles into various compartments of the cell [19]; transport of drugs and their slow release [20]; easy surface functionalisation; and the permeability and retention effect—EPR [19]. The C_60_ cage structure encapsulates heavy metals inside the fullerene and prevents them from leaking into the biological environment, which has been used in early cancer detection in diagnostic contrasts [19].

Many factors have been reported to regulate EMT, including growth factors [9], transcription factors [21] and surface stiffness [22]. In this work, we evaluated for the first time whether cell growth on the surface of fullerene C_60_ promotes MET and whether there is a negative correlation between the epithelial phenotype of cells and the growth factor cocktail-induced oxidative stress, inflammation and autophagy.

## 2. Materials and Methods

### 2.1. Characterisation of Fullerene Nanoparticles and Nanofilm

Fullerene C_60_ nanoparticles were purchased from SES Research (Houston, TX, USA) in the form of a dark-brown powder hardly dispersible in water. According to the manufacturer’s information, the C_60_ nanoparticles were produced using the arc discharge method. The C_60_ powder was suspended in Milli-Q ultrapure water at a concentration of 100 mg/L and then sonicated in an ultrasonic bath (Bandelin Electronic, Berlin, Germany). C_60_ hydrocolloid prepared in a volume of 5 µL was added onto Formvar/Carbon 200 Mesh, Cooper (Co. 201208, Agar Scientific, Essex, UK). The morphology of C_60_ nanoparticles was observed using transmission electron microscopy (JEM-1220 JEOL, Tokyo, Japan). Nanoparticle stability was measured in water, Dulbecco’s Modified Eagle Medium (DMEM, Gibco, Thermo Fisher Scientific, Waltham, MA, USA) and Roswell Park Memorial Institute 1640 (RPMI 1640) medium (Gibco, Thermo Fisher Scientific) using the Nano-ZS90 Zetasizer (Malvern Instruments, Malvern, UK).

To prepare the C_60_ nanofilm, the nanoparticle colloid was added to the bottom of a six-well polymer culture dish (10.4 µg/cm^2^ for 100 mg/L) and dried under sterile conditions. A heat cut of the dish wall was performed before placing the control sample (polymer dish) and test sample (polymer dish with nanofilm) in an atomic force microscope (AFM). The surface roughness parameters were determined using an AFM from Nanosurf (Liestal, Switzerland).

### 2.2. Cell Culture

The non-invasive liver cancer line HepG2 (HB-8065), the naturally invasive liver cancer line SNU-449 (CRL-2234) and the non-cancerous foetal fibroblast line HFFF2 (No. 86031405) were obtained from the American Type Culture Collection (Manassas, VA, USA) and the European Collection of Authenticated Cell Cultures (Salisbury, UK). Cell cultures were incubated at 37 °C in a humidified atmosphere of 5% carbon dioxide (CO_2_). HepG2 and HFFF2 cells were maintained in DMEM. SNU-449 cells were grown in RPMI 1640 medium. DMEM and RPMI 1640 medium were supplemented with 10% (*v*/*v*) foetal bovine serum (FBS), penicillin (100 U/mL) and streptomycin (100 mg/mL, Thermo Fisher Scientific). 

### 2.3. MTT Assay

The Cell Proliferation Kit I (MTT, No. M6494, Thermo Fisher Scientific) was used to determine the effect of fullerene nanofilm at concentrations of 50 (5.2 µg/cm^2^), 100 (10.4 µg/cm^2^), and 1000 mg/L (104 µg/cm^2^) on the viability of cells. Cells were counted and seeded on a 96-well plate at a concentration of 1.5 × 10^4^ cells per well. MTT stock solution at a concentration of 12 mM was prepared by adding 1 mL of sterile phosphate-buffered saline (PBS) to one 5 mg vial of MTT. Next, MTT stock solution was mixed and sonicated until it was dissolved. Then, 10 µL of MTT stock solution was added to a 100 µL culture medium. After 2 h incubation at 37 °C, the culture medium containing MTT stock solution was discarded. In the last step, cells were lysed with 100 µL/well of lysis buffer and incubated overnight. The lysis buffer contained triton ×100 (50 mL), isopropanol (45 mL), and HCl (two drops). The absorbance was measured at 570 nm using a Tecan Infinite 200 microplate reader (Tecan, Durham, NC, USA). The results of three replicates were averaged and expressed as % of control.

### 2.4. Anti-Inflammatory and Antioxidative Properties of Fullerene Nanofilm 

Cells were detached by trypsinisation and seeded on six-well plates coated or not coated with C_60_ nanofilm at a concentration of 1.5 × 10^5^ cells per well. Nanofilms were prepared as in Section 2.1. After incubation for 24 h, cells were collected by centrifugation at 1200 rpm and washed three times in PBS (pH 7.2). Protein from cell pellets was isolated using radioimmunoprecipitation assay buffer (RIPA, No. 8990, Thermo Fisher Scientific) with the addition of a protease and phosphatase inhibitor cocktail (No. 78442, Thermo Fisher Scientific) and using the TissueLyser LT instrument (Qiagen, Hilden, Germany). The homogenate was centrifuged at 6000 rpm for 30 min at 4 °C to remove cellular debris. In the supernatant containing the protein extract, the total protein concentration was determined in triplicate using the Bicinchoninic Acid Protein Assay Kit (BCA, No. 23225, Thermo Fisher Scientific) according to the manufacturer’s recommendations. The protein concentration was equalised in all samples based on the bovine serum albumin standard curve. All experiments were performed on protein lysates of the HFFF2, HepG2 and SNU-499 lines according to the scheme: (1) lysate isolated from cells growing on a polymer plate (control) and (2) lysate isolated from cells growing on a nanofilm-coated polymer plate. 

#### 2.4.1. Quantitative Detection of Transforming Growth Factor Using Enzyme-Linked Immunosorbent Assay

The inflammatory response of HFFF2, HepG2 and SNU-449 cells cultivated on C_60_ nanofilms was detected using the TGF beta1 Human ELISA Kit (No. BMS249-4, Thermo Fisher Scientific). Thirteen micrograms of protein isolated from cells was diluted to 200 µL with Assay Buffer. Then, 20 µL of 1 N HCl was added, and the solution was incubated for 1 h at room temperature. The pH was neutralised by adding 20 µL of 1 N NaOH. Samples prepared in this way were added to the plate (90 µL/well), followed by 10 µL of 1× Assay Buffer. TGF present in samples bound to antibodies adsorbed to the microwells. The negative control was 100 µL of 1× Assay Buffer. The plate was incubated overnight at 4 °C. All subsequent steps were performed according to the manufacturer’s instructions. The absorbance at 450 nm was measured using a Tecan Infinite 200 microplate reader (Tecan, Durham, NC, USA). The results of four replicates were averaged and expressed as % of control.

#### 2.4.2. Total Antioxidant Capacity Assay 

The OxiSelect™ Total Antioxidant Capacity Assay Kit (TAC, No. STA-360, Cell Biolabs, San Diego, CA, USA) was used to measure antioxidant enzymes; thus, it provides an indirect measurement of ROS. The assay quantifies the ability of an antioxidant to transfer one electron to reduce any compound, such as a free radicals, carbonyls and metals. Antioxidants neutralise radicals by transferring a single electron. Uric acid was used to create a standard curve in the range of 0.004–1 mM.

Twenty microlitres of diluted standards and 24 µg of samples were added to each well of a 96-well plate. Then 180 µL of 1× Reaction Buffer was added to each well and the initial absorbance at 490 nm was measured using a Tecan Infinite 200 microplate reader (Tecan). To initiate the reaction, 50 µL of the 1× Ion Reagent was added to each well. After incubation for 5 min, 50 µL of 1× Stop Solution was added to terminate the reaction. The absorbance at 490 nm was read again. The absorbance was calculated by subtracting the initial absorbance reading for samples and standards from the final readings for each. The standard curve was used to determine the millimolar uric acid equivalents (UAE). Finally, the results for samples were expressed as micromolar copper-reducing equivalents (CRE), which were calculated by multiplying UAE by 2189 μM Cu^2+^/mM uric acid. 

#### 2.4.3. Quantitative Detection of Glutathione Using Enzyme-Linked Immunosorbent Assay

A Glutathione ELISA Kit (No. Abx257145) was purchased from Abbexa (Cambridge, UK) and was used to determine the total glutathione (GSH) level in cell lysates. Cell lysates were diluted in PBS to a final concentration of 10 µg in 50 µL. To determine the GSH level, standards were prepared in the range from 1.56 to 100 µg/mL. Standards, test samples and a control (zero, diluent buffer) were added to the antibody-coated plate (50 µL/well in triplicate). Next, 50 µL of Detection Reagent A working solution was added to each well, and the plate was incubated for 45 min at 37 °C. After incubation, the wells were washed three times with 1× Wash Buffer. After the last wash, the liquid was completely aspirated from the wells. Then, 100 µL of Detection Reagent B working solution was added to each well, and the plate was incubated for 30 min at 37 °C. After incubation, the wells were washed again. In the last step, 90 µL of 3,3′,5,5′-tetramethylbenzidine (TMB Substrate) was added to each well, and the plate was incubated at 37 °C for 20 min. The reaction was stopped by adding 50 µL of Stop Solution. The absorbance at 450 nm was measured using a Tecan Infinite 200 microplate reader (Tecan). Changes in the GSH level in experimental group are expressed as % of control.

### 2.5. Mesenchymal–Epithelial Transition of Cells on Fullerene Nanofilm

Epithelial HepG2 and SNU-449 cells were cultured in DMEM and RPMI 1640 medium, respectively, with 2% FBS (*v*/*v*) and 20 ng/mL active growth factor cocktail, to induce the mesenchymal phenotype of the cells. The growth factor cocktail contained TGF-β1 (No. ab50036, Abcam, Cambridge, UK), TNF-α1 (No. PHC3015, Thermo Fisher Scientific) and EGF (No. PHG0315, Thermo Fisher Scientific) in a weight ratio of 1:1:1. This step is described in detail in a previous publication [9]. The HFFF2 cells, with a mesenchymal phenotype, served as a control; they were cultured in DMEM with 2% FBS (*v*/*v*) and were not treated with growth factors. Mesenchymal cells of the three cell lines were detached from culture bottles and transferred to C_60_ nanofilm-coated and non-coated culture wells. 

#### 2.5.1. Morphology of Epithelial and Mesenchymal Cells on C_60_ Nanofilm

Cell morphology was determined after incubation for 48 h on an ordinary plate (control) and a plate that had been coated with C_60_ nanofilm. The following cells were seeded on the plates: (1) HepG2 cells with an epithelial phenotype; (2) HepG2 with a mesenchymal phenotype after growth factor treatment; (3) SNU-449 cells with an epithelial phenotype; (4) SNU-449 cells with a mesenchymal phenotype after growth factor treatment; and (5) HFF2 cells with a mesenchymal phenotype. The morphology of cells grown on uncoated plate wells was compared with cells grown on C_60_ nanofilm-coated wells. The cells were stained with May Grünwald (No. 63590, Sigma-Aldrich, St. Louis, MO, USA) and Giemsa (No. 48900, Sigma-Aldrich) solutions. An inverted light microscope (Leica, TL-LED, Wetzlar, Germany) with a digital camera (Leica MC190 HD) and LAS V4.10 software (Leica) was used to take five photos for each group. 

#### 2.5.2. Two-Dimensional Invasion Assay

HepG2, SNU-449 and HFFF2 cell invasion was assessed using Culture-Insert 3 Well (No. 80366, Animalab, Germany). First, the C_60_ colloid was dried on the wells of a six-well plate in sterile conditions. Then, the inserts were placed in wells with or without a C_60_ nanofilm coating. The total area of the plate including the free gap and surface of the inserts was coated with carbon film. In the next step, epithelial (HepG2 and SNU-449) cells were seeded in the left well and mesenchymal cells (all lines) were seeded in the right well of the culture insert to achieve a monolayer. In the central well, an imitation of a blood vessel was created; it comprised human umbilical vein endothelial cells (HUVECs, No. C0155C, Thermo Fisher Scientific) and Engelbreth–Holm–Swarm (EHS) Matrix Extract (No. 125-2.5, Cell Applications Inc, San Diego, CA, USA). HUVECs were maintained in Human Large Vessel Endothelial Cell Basal Medium without phenol red (Medium 200, M200PRF500, Thermo Fisher Scientific) and mixed with EHS Matrix Extract in a 1:1 ratio. Twenty microliters of this mixture was added to the centre well. After incubation for 24 h, the inserts were removed, and the cell layer was washed with PBS. The wells were filled with 2 mL of the appropriate medium for the cell line supplemented with 2% FBS (*v*/*v*). Using the LAS V4.10 software (Leica, Wetzlar, Germany), photographs were taken after incubation for 48 h at 2.5× magnification using a digital camera (Leica MC190 HD) mounted on an inverted microscope. 

#### 2.5.3. Cell Invasion through the Extracellular Matrix

Cell Invasion Assays (No. ECM555 and ECM550, Sigma-Aldrich) were used to study the invasion of tumour cells through a basement membrane matrix of proteins derived from the EHS mouse tumour. The inserts contained a polycarbonate membrane with 8 µm pores. In these experiments, inserts coated with EHS Matrix Extract were modified by adding 10.4 µg/cm^2^ of C_60_ colloid or water (control); the inserts were allowed to dry overnight. Cells before and after treatment with growth factors (as described in Section 2.5) were seeded at a concentration of 1 × 10^5^ (No. ECM555) or 3.0 × 10^5^ (No. ECM550) on the modified insert in triplicate. Cells were maintained in a serum-free medium. In the lower chamber, 500 µL of medium containing 10% FBS (*v*/*v*) or HUVECs in Medium 200 was added. 

For fluorescence detection, after 24 h (No. ECM555), invading cells were detached and stained with CyQuant GR Dye according to the ECM555 protocol. The fluorescence at 480/520 nm was measured using a Tecan Infinite 200 microplate reader (Tecan).

For absorbance detection, after 72 h (No. ECM550), non-invading cells and ECM gel were removed using a cotton-tipped swab. Then, 500 µL of crystal violet was added to new wells, and the inserts were transferred to the dye. After 20 min, the inserts were washed several times with a large volume of water. The cells were observed using an inverted light microscope (Leica) and then dissolved using 10% acetic acid. The absorbance at 560 nm was measured using a Tecan Infinite 200 microplate reader (Tecan).

#### 2.5.4. Metalloproteinase Expression Assay

The Human MMP Antibody Array (No. ab134004, Abcam) was used to determine the levels of 10 targets, namely MMPs and TIMPs. Mesenchymal cancer cells obtained as described in Section 2.5 were seeded on coated and uncoated six-well plates at 1.5 × 10^5^ cells per well. After incubation for 48 h, the cells were collected by centrifugation at 1200 rpm and lysates using RIPA buffer with the addition of a protease and phosphatase inhibitor cocktail. The protein concentration of the extract was measured using the BCA method (No. 23225, Thermo Fisher Scientific). Then, 200 µg of protein was added to each blocked protein membrane and incubated overnight. The next steps were performed according to the manufacturer’s instructions. Chemiluminescence detection was performed using Azure c400 (Azure Biosystems, Dublin, CA, USA). The ImageJ^®^ 1.54dt software (National Institutes of Health, Bethesda, MD, USA) was used for quantification.

#### 2.5.5. Expression of Mesenchymal Markers Using Western Blotting 

The protein isolated as described in Section 2.5.4. was used to determine the expression of mesenchymal markers by Western blot. The protein concentration of the extract was determined with the BCA method. A sample buffer containing β-mercaptoethanol (Bio-Rad Laboratories, Munich, Germany) was added to the protein extract at a ratio of 1:5. After denaturation, equal amounts of protein from each sample (50 µg) were loaded onto a 7.5% or 12% polyacrylamide gel (TGX Stain-Free™ FastCast™ Acrylamide Starter Kit, No. 1610180 and 1610185, Bio-Rad Laboratories). Electrophoresis was run initially at 90 V and then at 150 V for 1.5 h in 25 mM Tris–glycine–sodium dodecyl sulphate buffer. After electrophoresis, the separated protein was transferred to polyvinylidene difluoride membranes with a Trans-Blot Turbo Transfer System (Bio-Rad Laboratories). The membranes were blocked using an iBind™ Flex Solution Kit (No. SLF2020, Thermo Fisher Scientific). The blocked membranes were labelled with the primary antibody (Table 1), washed and then labelled with goat anti-rabbit IgG (ab97048, Abcam) or goat anti-mouse IgG (ab97020, Abcam) using an iBind™ Flex Western Device (No. SLF2000, Thermo Fisher Scientific). The protein bands were visualised using Azure c400 (Azure Biosystems). Image analysis was performed using the ImageJ^®^ 1.54d software (National Institutes of Health).

#### 2.5.6. HepG2 Cell Proteome Analysis Using Mass Spectrometry

HepG2 mesenchymal cells obtained after 48 h treatment with 20 ng/mL of a growth factor cocktail were detached with trypsin and transferred to ordinary plates and C_60_ nanofilm-coated plates. After incubation for 48 h, the cells were detached and harvested by centrifugation at 1200 rpm and then washed several times with PBS. Samples prepared in this way were used for proteomic analysis using the Mini MS Sample Prep Kit (No. A40006, Thermo Fisher Scientific) according to the manufacturer’s instructions. The concentration of the isolated protein was measured spectrophotometrically using the Pierce™ 660 nm Protein Assay Kit (No. 22662, Thermo Fisher Scientific). A volume corresponding to 100 pg of total protein was taken for analysis. This protein was subjected to a reduction of sulphide bridges, the alkylation of free sulfhydryl groups, and enzymatic hydrolysis using trypsin and LysC. The resulting peptide mixtures were purified using the solid phase extraction technique on C18 columns. The purified samples were evaporated in a vacuum concentrator and suspended in water (liquid chromatography–mass spectrometry [LC-MS] grade) containing 0.1% formic acid. The Pierce Quantitative Fluorometric Peptide Assay kit (No. 23290, Thermo Fisher Scientific) was used for the fluorometric measurement of the total peptide concentration. The peptide concentration was then adjusted to 0.8 pg/µL in all samples. To monitor the operating conditions of the measurement system, quality control samples were prepared by mixing 5 µL of the control sample and 5 µL of the test sample. Measurements were made using an Orbitrap Exploris 480 mass spectrometer (Thermo Fisher Scientific) and a Neo Vanquish liquid chromatograph (Thermo Fisher Scientific). The proteins were identified based on a SwissProt Homo Sapiens database search. Statistical analysis (volcano plots) was performed using the Proteome Discoverer 3.0 software (Thermo Fisher Scientific). The interaction network of protein and protein classes was created using the ZS Revelen website (Intomics, Denmark) and Panther Classification System 18.0 software.

### 2.6. Statistical Analysis 

The results were analysed by unpaired *t*-test and one-way analysis of variance (ANOVA) with Bonferroni’s multiple comparisons test (for MTT assay) using GraphPad Prism 8 software version 8.0.2 (San Diego, CA, USA) and Proteome Discoverer 3.0 (Thermo Fisher Scientific). Statistical significance is indicated by asterisks: * *p*  <  0.05, ** *p*  <  0.01 and *** *p*  <  0.001.

## 3. Results

### 3.1. Morphology and Stability of Fullerene Nanoparticles

The fullerene C_60_ nanoparticles had irregular spherical shapes and a size range from about 12 to 300 nm (Figure 1A,B). The fullerene nanoparticles were hydrophobic and showed poor suspension in water. Colloids of nanoparticles showed a high dispersion stability of −25 ± 3.7 mV. The charge of fullerene was much lower after suspension in culture media: −8.8 ± 1.0 mV (Figure 1C).

We examined the roughness of a standard polystyrene plate and the same plate coated with fullerene colloid by using an AFM. The uncoated plate had an average roughness of 7.2 nm. The applied nanoparticles filled the irregularities of the uncoated plate, which resulted in a lower average roughness of 2.6 nm (Figure 1D,E). Moreover, C_60_ created a thread-like surface as a result of the self-assembly of circular structures.

### 3.2. The Effect of Concentrations of C_60_ Nanofilm on Cytotoxicity

The effect of different concentrations of C_60_ colloids forming nanofilms on the viability of HepG2, SNU-449, and HFF2 cells was examined using the MTT (3-(4,5-dimethylthiazol-2-yl)-2,5-diphenyltetrazolium bromide) assay. The nanofilm C_60_, at concentrations of 50 and 100 mg/L, resulted in no changes in cancer cell activity (Figure 2). The C_60_ 1000 mg/L nanofilm significantly reduced the viability of the HepG2 cell line by 21.2% (*p* = 0.019). All experimental factors were toxic for HFF2 normal cells. However, HFF2 cell viability was strongest inhibited on nanofilm C_60_ at a concentration of 1000 mg/L. These results allowed us to demonstrate the biocompatibility of the C_60_ nanofilm at concentrations of 50 and 100 mg/L. A concentration of 100 mg/L was selected for further studies.

### 3.3. The Effect of C_60_ Nanofilm on Transforming Growth Factor Beta 1 Synthesis and Oxidant Levels

The main cytokine involved in EMT induction is TGF-β1; thus, we investigated the effect of fullerene nanofilm on the secretion of this factor by cells. After culturing HepG2 cancer cells on C_60_ nanofilm, we observed a trend for reduced TGF-β1 expression, although the change was not significant. We noted a similar, but significant, effect for SNU-449 cells: an 18% reduction in TGF-β1 expression. The C_60_ nanofilm did not affect TGF-β1 secretion in the normal HFF2 fibroblast line (Figure 3A).

Oxidative stress is one of the causes of cellular damage and also promotes EMT. Thus, we evaluated the effect of C_60_ nanofilm on the total antioxidant content and the GSH level, which protect against oxidative stress in the cell. The C_60_ nanofilm had no significant effect on the total antioxidant content in the two tested cancer cell lines (Figure 3B). However, the antioxidant capacity was increased in HFF2 cells grown on the C_60_ nanofilm. GSH is a natural antioxidant and a marker of cell redox homeostasis. Growing HepG2 and HFF2 cells on the C_60_ nanofilm did not affect the GSH level (Figure 3C). However, there was a slight, but significant, 6% increase in GSH for SNU-449 cells cultivated on the C_60_ nanofilm compared with the control.

### 3.4. The Effect of C_60_ Nanofilm on Cell Morphology and Two-Dimensional Invasion towards Endothelial Cells

Currently, little is known about the metastasis of tumour cells from the primary site of malignancy to neighbouring stromal tissue or distant localities. This process is often connected with signals from endothelial cells, which are a natural source of signals and growth factors for liver cancer cells in co-culture. Therefore, we performed a co-culture to provide a source of chemo-attractants. A two-dimensional migration assay provides information on the ability of cells to migrate into a cell-free area. However, this assay does not reflect the real conditions of tumour cell invasion through the vascular basement membrane. Hence, we used a co-culture of cytokine-induced (mesenchymal phenotype) and non-induced (epithelial phenotype) liver cancer cells. We added HUVECs suspended in ECM components such as laminin, type IV collagen and growth factors between cells with epithelial and mesenchymal phenotypes (Figure 4A). We used this assay to evaluate the effect of the C_60_ nanofilm on cell adhesion and migration.

On the uncoated plates, HepG2 and SNU-449 cancer cells with the invasive mesenchymal phenotype had a faster rate of growth and migration and impaired cell adhesion compared with the same cells with the epithelial phenotype. Moreover, HepG2 cells with the mesenchymal phenotype detached from the uncoated plate. However, on the C_60_ nanofilm, HepG2 cells with the mesenchymal and epithelial phenotypes showed an increase in cell–cell and cell–ECM junctions, denoted as large and tight cell clusters (Figure 4B, Appendix A).

The C_60_ nanofilm was also biocompatible with SNU-449 cells. Of note, SNU-449 cells with the epithelial phenotype did not change their morphology after cultivation on C_60_ nanofilm. The width of the cell-free scratch was smaller on the C_60_ nanofilm-coated plate compared with the uncoated plate. We observed a different effect for SNU-449 cells with the mesenchymal phenotype: the C_60_ nanofilm enhanced cell–cell interactions and there were more nuclei in the field of view but occupying less area compared with the control. The results confirmed that C_60_ nanofilm attenuated the invasion of cancer cells with the mesenchymal phenotype and favoured the recovery of the epithelial phenotype (Figure 4C). 

We used human Caucasian foetal foreskin fibroblast HFF2 cells as control cells with a mesenchymal phenotype. They showed the fastest migration rate among the tested cell lines. The HFF2 cell morphology did not change after growth on C_60_ nanofilm. The cells also showed a similar rate of migration regardless of the area they inhabited (Figure 4D).

### 3.5. The Effect of C_60_ Nanofilm on Three-Dimensional Invasion

In an in vivo environment, cells invade through the basement membrane and migrate towards endothelial cells. Thus, we optimised the three-dimensional invasion of cells suspended in serum-free medium by using two sources of chemo-attractants, namely HUVECs in Medium 200 and culture media with FBS. We found that cell invasion was independent of the chemo-attractant used; therefore, we used culture media with FBS for the subsequent experiments (Figure 5A).

In the next step, we examined the effect of the C_60_ nanofilm covering the basement membrane proteins on the migration of epithelial and mesenchymal cells towards chemo-attractants. C_60_ nanofilm reduced the invasive capacity of SNU-449 cells with the mesenchymal phenotype but did not affect the invasive capacity of SNU-449 cells with the epithelial phenotype and the remaining cell lines after incubation for 24 h (Figure 5B). There were similar results for SNU-449 cells after incubation for 72 h. However, control HFF2 cells showed increased invasion on C_60_ nanofilm-coated basement membrane proteins after incubation for 72 h (Figure 5C).

### 3.6. The Effect of C_60_ Nanofilm on Degradation of the Extracellular Matrix

A balance between the synthesis and degradation of ECM proteins is essential to maintain the resting state of cells. Thus, we evaluated the expression of seven MMPs and three TIMPs. The C_60_ nanofilm had the greatest effect on reducing the expression of TIMPs in cancer cells. HepG2 cells cultured on C_60_ nanofilm showed a reduced expression of TIMP-1, TIMP-2, TIMP-4 and MMP-1. In SNU-449 cells, C_60_ nanofilm downregulated the expression of TIMP-1, TIMP-2, TIMP-4, MMP-1 and MMP-9 (Figure 6A,B, Appendix A).

### 3.7. The Effect of C_60_ Nanofilm on Key Proteins Involved in the Mesenchymal–Epithelial Transition and Oxidative Stress

We used Western blotting to determine whether changes in invasion and MMP activity were associated with key EMT markers (Figure 6C). The C_60_ nanofilm increased the expression of beta-catenin, snail/slug and Smad4 in HepG2 cells compared with the control group. Even with cytokine treatment, we could not detect vimentin in HepG2 cells. SNU-449 cells had a low total beta-catenin content, below the level of detection. Growing SNU-449 cells on C_60_ nanofilm did not alter the expression of vimentin and snail/slug. However, mechano-signalling from the C_60_ nanofilm resulted in a very marked increase in Smad4 expression in SNU-449 cells. Given the strong correlation between oxidative stress, inflammation and EMT, we analysed the expression of catalase, a key antioxidant. HepG2 and SNU-449 cells grown on the C_60_ nanofilm showed increased catalase expression. 

### 3.8. The Effect of C_60_ Nanofilm on the Proteomic Profile of HepG2 Cells

We examined the overall proteomic profile of HepG2 cells grown and not grown on C_60_ nanofilm to identify differences in the levels of proteins involved directly or indirectly in phenotype transition. We identified a total of 6295 proteins in the HepG2 cell lysates. Among all HepG2 cell proteins, as many as 96 proteins showed significantly altered levels under the influence of growth on C_60_ nanofilm compared with the uncoated control plate (fold change ≥ 1.4). The quantitative method showed that most of the altered proteins were downregulated (50) by growth on the C_60_ nanofilm (Figure 7A). According to the Panther Classification 18.0 software, the most abundant protein classes were metabolite interconversion enzymes (10.5%), calcium-binding proteins (8.4%) and cytoskeletal proteins (8.4%) (Figure 7B). We performed principal component analysis (PCA) based on protein count values from LC-MS/MS. Principal components 1 and 2 (PC1 and PC2, respectively) explained 73.4% of data variability (49.0% for PC1 and 24.4% for PC2) among six samples belonging to two study groups (Figure 8).

We selected 41 proteins involved in EMT as well as autophagy, inflammation, oxidative stress, migration and pathways involved in insulin metabolism and proliferation for the next step. An analysis of heat maps showed that the C_60_ nanofilm most strongly reduced the expression of keratin 28, 27 and 25 by about 3.5-fold (Figure 9A). In addition, the C_60_ nanofilm regulated as many as eight proteins belonging to the keratin family. The proteins with the highest degree of interaction with other proteins included those from the keratin family, actin, ribosomal protein S6 kinase beta-2, and exocyst complex component 3 (EXOC3). Physical protein–protein interactions are shown in Figure 9B.

## 4. Discussion

We included two panels of experiments to determine whether surface modification by applying C_60_ nanofilm could restore the epithelial (dormant) phenotype of liver cancer cells. The first panel of experiments concerned characterising C_60_ nanofilm in terms of physicochemistry and its toxicity and anti-inflammatory and antioxidant properties. The second panel of experiments involved inducing the invasive, mesenchymal phenotype of liver cancer cells by using a growth factor cocktail and then analysing whether C_60_ nanofilm could reduce the invasiveness of these cells.

The toxicity of fullerene depends on its size and surface chemistry, dose, exposure time, extracellular or intracellular exposure, and the presence of tetrahydrofuran after its synthesis [23]. The supply of oxygen in the tumour is also important, which determines the effectiveness of fullerene as a photosensitizer [24]. C_60_ nanofilm was toxic to cancer and normal cells at a concentration of 1000 mg/L. However, the use of a thinner nanofilm layer (100 mg/L) did not affect the growth of cancer cells and may even protect the liver against painkillers, as previously demonstrated by Kuznietsova et al. [25]. For this reason, the nanofilm at a concentration of 100 mg/L was used in further research. It seems that the structure of fullerenes determines their toxicity, and carbon soot C_60_/_70_ with a predominance of C_70_, even at low concentrations, induces the necrosis of mouse fibroblast cells L929, rat glioma C6, human glioma U251 [26] and human lung cancer A549 [27]. Cell death after exposure to high doses of C_60_ fullerene may be associated with ROS-dependent cell membrane damage, as previously described for melanoma cells B16 [28]. HepG2 cells show an IC_50_ of 77.9 µg/mL for C_60_ fullerene colloids added to the culture fluid [12]. Nanofilm strongly bound to the surface of culture plates is more difficult to be taken up by cells and therefore only shows cytostatic properties, without any clear cytotoxicity at much higher concentrations.

EMT is closely related to inflammation, which is caused by the generation of oxygen-free radicals. There is a strong feedback loop between these three processes. For this reason, our next step was to determine whether the C_60_ nanofilm inhibits the secretion of cytokines and has an antioxidant effect against HepG2 and SNU-449 cancer cells with the epithelial phenotype and foetal HFF2 cells with the mesenchymal phenotype. The cytokine TGF-β1 is widely recognised as an inducer of EMT and a pro-inflammatory factor associated with HCC [29]. Zhu et al. [30] reported that 10 mg/L of graphene oxide increased the expression of the TGF receptor. We showed that C_60_ nanofilm as a growth surface reduced the intracellular synthesis of TGF-β1 in SNU-449 cells but did not affect this cytokine in the other two cell lines. Knowing that an increased level of TGF-β1 accompanies fibrotic diseases, overproduction of ROS in mitochondria and deregulation of redox balance, we assumed that it would also affect the level of antioxidants [31]. Studies have shown that fullerene C_60_ enhanced the antioxidant capacity of HFF2 and SNU-449 cells by increasing the total content of antioxidants and GSH. Namadr et al. [32] reported similar results: the oral administration of fullerenes to rats increased the activity of catalase and superoxide dismutase (SOD) in the liver. Vani et al. [33] also showed that C_60_(OH)_18–22_ polyhydroxylated fullerene nanoparticles increase the GSH and SOD levels in the brain of ischaemic rats. GSH is a tripeptide comprising cysteine, glycine and glutamate, and 90% of GSH is produced in the liver. Thus, the deregulation of the GSH level may have effects throughout the body. Patients with liver disease and diabetes have reduced serum GSH levels [34]. In addition, fullerene reacts with oxygen free radicals and mimics the activity of cellular antioxidants and protects against lipid peroxidation, protein modification and DNA damage [35].

The suppression of metastasis is a major concern in recurrence after liver resection. The invasive phenotype is a temporary, reversible and difficult-to-detect phenotype. Therefore, in the next stage of the experimental work, we induced the invasive phenotype of HepG2 and SNU-449 by using a cocktail we developed in a previous study; it contains TGF-β1, TNF and EGF [9]. However, the in vitro culture of mesenchymal tumour cells is limited by the lack of cancer-related cells, tissues and organs, and in particular the imitation blood vessels into which cells migrate during metastasis. Therefore, to determine the effect of C_60_ nanofilm on two- and three-dimensional invasion, we co-cultured liver cancer cells with HUVECs, which served as the migratory target of actively invasive cancer cells. The results confirmed that HepG2 and SNU-449 cells with the invasive, mesenchymal phenotype gained the ability to actively migrate by weakening the cell–cell and cell–ECM connections compared with the same cells with the epithelial phenotype. In experiments using the modified cells, we proved that the C_60_ nanofilm inhibited the detachment of invasive HepG2 cells from the polystyrene plate and promoted their adhesion to the modified surface. The growth of invasive SNU-449 cells on the C_60_ nanofilm resulted in the formation of strong intercellular connections, which we observed as less cell scattering and slower migration towards endothelial cells compared with the control cells. In similar experiments, other researchers have shown that fullerene derivatives reduced the wound closure of breast cancer stem cells [36] and glioblastoma multiforme [37]. We confirmed the obtained results by using a Boyden chamber that mimics the invasion of cells through the reconstructed basement membrane. We showed that the nanofilm inhibited the three-dimensional invasion of SNU-449 cells with the induced invasive, mesenchymal phenotype, but it did not affect the same cells with the epithelial phenotype. This effect can be explained by the anti-inflammatory and antioxidant properties of fullerene, which are only relevant when the redox balance is disturbed, especially in a cytokine storm. Moreover, the observed phenomenon can be explained by the presence of hepatitis B virus (HBV) DNA sequences (grade II–III, diffusely spreading cells) in naturally invasive cell lines such as SNU-449. According to the literature, HepG2 cells are considered non-invasive with low vimentin expression [38], so their growth on nanofilm, even in an inflammatory environment, may be different. Lucafo et al. [39] demonstrated that the fullerene-mediated inhibition of invasion is independent of the integrin pathway. In their work, 25 µM of C_60+_ (a cationic form of fullerene) inhibited the invasion of HT-29 colorectal cancer cells by 75% after incubation for 96 h. The charge on the surface of the fullerenes tested by the authors was negative both in water and in the culture medium. The charge on the surface of fullerene and even the chemistry of the surface does not determine the invasiveness of the cells; the morphology of the surface, its geometry and the size of the nanostructures that form it are the key factors [40]. Precise surface control using photolithography and electrodeposition showed that nanostructures in the size range of 100–500 nm restored the epithelial phenotype of invasive breast and prostate cancer cells for at least 7 days, even after the removal of the nanotopographic stimulus. Much larger structures (2000 nm) did not promote MET [40]. Studies using transmission electron microscopy have shown that fullerene crystals that promoted the dormant, epithelial phenotype served as topographic stimuli and were 12–300 nm in size. The surfaces of reduced graphene oxide and graphene oxide with a roughness of 1 nm triggered EMT in A549 lung epithelial cells, PC3 prostate cancer cells and HepG2 liver cancer cells [30,41]. The use of a higher concentration of graphene oxide reversed this effect probably by increasing the number of graphene layers and increasing the extremely low surface roughness to a few nanometres, which promoted cell adhesion [30] and resembled the roughness of surfaces made using fullerene (2.6 nm).

In fibrotic diseases such as HCC, ECM remodelling is dysregulated by the upregulation of MMPs. A disturbed balance between MMPs and TIMPs leads to inflammation, ECM degradation, invasion, and escape from apoptosis. According to Xu et al. [42], the upregulation of MMP-1, MMP-9, MMP-12 and MMP-14 correlates with a poor survival prognosis. The reduced expression of MMP-1 (collagenase) and MMP-9 (gelatinase B) confirms the role of fullerene in inhibiting excessive cell migration through the ECM. MMP-1 is considered the main marker of success of immunotherapy in liver cancer [42]. Of note, MMP-1 was inhibited in both cancer cell lines grown on the C_60_ nanofilm. We expected that the inhibition of MMP expression would be due to an increase in the expression of enzymatic MMP inhibitors; however, it turned out that TIMPs have other independent biological functions and may act independently of MMPs [43]. Fullerene affected the downregulation of TIMP-1, TIMP-2 and TIMP-4. In clinical studies, the overexpression of TIMP-1 and TIMP-4 has been associated with a poorer prognosis in many tumours; they activate signalling pathways related to inflammation and fibrosis—for example, mitogen-activated protein kinase (MAPK) pathways [44]. Wójcik et al. [45] showed, similarly to our study, a downregulation of TIMP-2 expression after adding diamond and graphene oxide at a concentration of 50 mg/L to the pancreatic adenocarcinoma cell lines AsPC-1 and BxPC-3. According to Kai et al. [46], reduced TIMP-2 expression induces the formation of invadopodia in HCC cells and correlates with the invasion of tumour cells into adjacent sites in the liver parenchyma and even with lung metastases under hypoxic conditions. In the present study, we did not observe such an effect after TIMP-2 expression had been reduced, perhaps due to the lack of a direct effect on the change in MMP-2 expression [47].

To understand the mechanisms by which fullerene influences the inhibition of invasion, we examined the β-catenin → vimentin → Smad4 → snail/slug signalling pathway. Molecular markers of EMT include a reduced expression of the β-catenin/E-cadherin complex and an increased expression of vimentin and snail/slug transcription factors [48]. We showed that the C_60_ nanofilm affected the HepG2 and SNU-449 cancer lines differently. In HepG2 cells, the C_60_ nanofilm led to an overexpression of β-catenin, snail/slug and Smad4 at the protein level. In SNU-449 cells, however, the C_60_ nanofilm only increased Smad4. The increase in β-catenin in HepG2 cells indicates stronger cell–cell connections compared with the control. It seems that the C_60_ nanofilm caused partial EMT reversion in HepG2 cells, that is, the formation of a hybrid phenotype that is often associated with E-cadherin inhibition by the snail/slug transcription factors [5]. After cultivation on C_60_ nanofilm, both cell lines showed an overexpression of the main signalling transmitter TGF-β1–Smad4. Based on the literature, the role of Smad4 is unclear: researchers have reported its participation in HCC and bone metastases as well as the suppression of other types of cancer, such as colorectal and pancreatic cancer [49]. However, the role in hepatoma is not clear due to the fact that Smad4 overexpression has been reported in patients with extrahepatic cholangiocarcinoma, while reduced expression has been found in patients with intrahepatic cholangiocarcinoma. These differences may be due to the heterogeneity of HCC [50]. The hyperactivation of TGF/Smad2,3 together with an increase in Smad4 expression, promotes EMT. However, in our work, C_60_ nanofilm decreased TGF secretion in both tumour cell lines, especially SNU-449 cells. In addition to its role in TGF signalling, Smad4 increases immune cell infiltration, which makes the tumour more sensitive to therapy [49]. In hepatoma, the activation of the p53/Smad4 pathway is also associated with apoptosis [51].

The first panel of experiments highlighted the antioxidant properties of C_60_ nanofilm against epithelial cells. ROS are produced in physiological processes but also as a response to chronic inflammation. According to Liu et al. [36], TGF increases the amount of ROS and inhibits the expression of cellular antioxidants. Knowing that there is a feedback loop between TGF and ROS, we confirmed the effect of C_60_ nanofilm on the redox state of mesenchymal cancer cells in an inflammatory environment. We used the main reducer of hydrogen peroxide to oxygen and water as a marker: catalase. The C_60_ nanofilm increased the expression of catalase in peroxisomes, in particular in HepG2 cells. Grebowski et al. [52] reported similar results: hydroxyfullerene C_60_(OH)_36_ at a concentration of 50–150 mg/L increased catalase expression by 24% in human erythrocytes incubated for 3 and 48 h and subjected to oxidative stress. The authors explained that the mechanism of action of the fullerene derivative involves a conformational change of catalase, which protects the -SH groups against ROS-induced oxidation. In another study, a high ratio of catalase expression in liver tumour samples relative to healthy tissue meant a better prognosis for patients and reduced the risk of metastasis [53].

HCC is a consequence of inflammation that induces a mesenchymal phenotype. The use of the naturally mesenchymal line SNU-449 with low E-cadherin expression and high vimentin expression will not provide us with full information about the effect of C_60_ on the reversal of cytokine-induced EMT. To better understand EMT induced by cytokines and MET generated by growth on C_60_ nanofilm, we studied the proteome of HepG2 cells. The largest group of proteins with altered levels in the cell lysate were proteins of intermediate fibres: keratins. Decreased levels of keratins are associated with the weakening of intercellular connections such as desmosomes and tight junctions, while their overexpression is associated with liver damage and fibrosis [31,54]. The knockout of single genes encoding keratin 6 [55], 19 [56], 8 and 18 [57]—and not the knockout of all genes involved in keratin synthesis—is associated with an aggressive cell phenotype in lung adenocarcinoma, HCC, cervical cancer and breast cancer. C_60_ nanofilm regulated the level of keratins that have not been well characterised: it decreased keratin 1, 2, 10, 25, 27 and 28, and increased keratin 9 and 12. These proteins are probably not involved in promoting EMT. Kuznetsova et al. [12] showed that fullerene C_60_ inhibited the expression of pan-cytokeratins in HepG2 cells, a phenomenon that was responsible for improving the condition of the liver in acute and chronic cholangitis. These findings support the beneficial effect of C_60_ nanofilm.

In the cell nucleus, fullerene downregulated the expression of Smad nuclear-interacting protein 1 (SNIP1). According to the literature, SNIP1 interacts with Smad4 and suppresses TGF-β/Smad signalling [58]. Our results confirm that the decrease in SNIP1 expression enhances Smad4 protein expression in HepG2 cells grown on C_60_ nanofilm. However, it is controversial that despite the overexpression of most components of the TGF-β/Smad pathway, two- and three-dimensional invasion was inhibited. For this reason, we sought other causes of reduced invasion in the processes affecting EMT, namely autophagy, inflammation and oxidative stress.

Autophagy is a lysosomal degradation pathway in which cells exposed to starvation or hypoxia form autophagosomes and digest their own macromolecules to maintain homeostasis. Autophagy allows cancer cells to survive under metabolic stress and facilitates EMT. In this and previous work, we have shown that the EMT of HCC cells occurs in a medium with a low FBS content (2% *v*/*v*), indicating a strong relationship between EMT and autophagy [9]. Studies using MS have shown that fullerene regulated proteins responsible for both the formation of autophagosomes WDR45B [59] and TMEM59 [60] as well as TTC5 autophagy inhibition [61]. However, the greatest difference in expression was noted for the transmembrane protein TMEM59. This protein promotes autophagic flux by activating microtubule-associated protein light chain 3 (LC3) and is an inducer of excessive autophagy leading to cell death [60]. It follows that the C_60_ nanofilm inhibited autophagy by reducing the level of the TMEM59 protein and thus limited the supply of nutrients needed for excessive cell proliferation in advanced cancer stages [62]. Knowing that EMT and cell invasion occur more strongly under starvation conditions, the reduction in autophagy markers due to fullerene probably restores the resting phenotype of cells. Dash et al. [63] showed that TGF-β induces autophagosome formation in human hepatoma. We found that the C_60_ nanofilm inhibited the synthesis of this cytokine in liver cancer cell lines.

Nutrient deficiency due to the uncontrolled proliferation of tumour cells enhances autophagy and is accompanied by insulin resistance. Accumulating evidence indicates that the poor prognosis of liver cancer is associated with type 2 diabetes and the blockage of insulin signalling pathways in hepatocytes. According to Li et al. [64], the inhibition of autophagy may reduce resistance to chemotherapeutic agents. Based on proteomic analysis, HepG2 insulin-resistant hepatoma cells grown on C_60_ nanofilm had altered levels of proteins involved in the transport of glucose transporter type 4 (GLUT4) to the plasma membrane of hepatocytes. Fullerene nanofilm, acting via an overexpression of EXOC3 and G kinase-anchoring protein 1 (GKAP1), increased GLUT4 exocytosis, glucose uptake in hepatocytes and glucose storage as glycogen [65]. According to Fujimoto et al. [66], the overexpression of EXOC3 in adipocytes and skeletal muscle increases glucose transport in response to insulin. Disorders of GLUT4 translocation are associated with insulin resistance and non-insulin-dependent diabetes mellitus in humans. Therefore, it seems that fullerene can restore normal glucose uptake in liver tissue.

Based on quantitative proteomics using MS, we found that the C_60_ nanofilm changed the expression of as many as 11 proteins identified as potential regulators of inflammation. Non-alcoholic steatohepatitis (NASH) leads to liver damage, inflammation and fibrosis due to the presence of excessive amounts of triglycerides in hepatocytes. An abnormal expression of the nuclear receptor NR2F6 disrupts the balance of fatty acid uptake, synthesis and transport [67]. The C_60_ nanofilm increased the expression of NR2F6, which, according to the literature, correlates with NASH and intensifies the production of cytokines. On the other hand, the C_60_ nanofilm also upregulated the transmembrane plasminogen receptor, a deficiency of which in mice fed a high-fat diet caused weight gain, fatty liver and insulin resistance [68]. Fullerene appears to have an anti-inflammatory effect by targeting the nuclear factor κB (NF-κB) transcription factor pathway, which is responsible for inflammation-induced cell death [69]. The C_60_ nanofilm regulated three proteins involved in the NF-κB pathway, namely TNIP1 (upregulation), RPS6KB2 (downregulation) [70] and CCDC22 (upregulation) [71]. TNIP1 is a repressor of pro-inflammatory and oncogenic NF-κB signalling [72]. Moreover, TNIP is rapidly degraded during autophagy and in the early stages of inflammation, confirming previous findings that fullerene inhibits autophagy and has anti-inflammatory properties [69]. The RPS6KB2 protein kinase is activated by various growth factors, including EGF and platelet-derived growth factor, and insulin. According to the literature, the reduction of RPS6KB2 expression by fullerene includes the reduction of NF-κB activation by angiotensin II [70]. C_60_ nanofilm also downregulated cytidine deaminase apolipoprotein (APOBEC3A, or A3A), whose main function is the deamination of cytosine to uracil during mutagenesis and viral infections. While in viral infection the role of APOBEC3A is based on the innate immune response and mutations in the viral genome that inhibits their replication, in cancers, APOBEC is the cause of their mutations, the heterogeneity and diversity of tumours, and drug resistance [73]. Knowing that APOBEC3A expression is induced by interferon and inflammatory signals [74], we can conclude that fullerene suppressed inflammation and reduced the aggressiveness of HCC by reducing chromosomal instability in vitro. Another protein downregulated by the C_60_ nanofilm was the protein acyltransferase ZDHHC20, involved in protein palmitoylation. Recent data have revealed elevated expressions of ZDHHC17, ZDHHC18, ZDHHC19 and ZDHHC20 in the tissues of patients with kidney renal clear cell carcinoma; moreover, ZDHHC20 may regulate the immune status of patients [75]. The C_60_ nanofilm also downregulated ecto-5’-nucleotidase (CD73, the NT5E gene), thereby reducing the conversion of adenosine 5’-monophosphate to adenosine. In malignant tumours, CD73 overexpression is responsible for creating an immunosuppressive tumour microenvironment, is a suppressor of T lymphocytes, and promotes cancer cell invasion and EMT [76]. High levels of CD73 have been shown during fibrosis, steatosis and drug-induced liver damage [77].

C_60_ nanofilm inhibited oxidative stress generated in cancer cells by treating them with a growth factor cocktail due to the increased synthesis of antioxidant enzymes such as catalase in the cells. The state of redox balance in the cell, however, is the result of the interaction of various anti- and pro-oxidants. In the last step of the work, we wanted to evaluate the effect of fullerene on other antioxidants involved in redox homeostasis, such as metallothionein 1X and 2A (MT1X and MT2A, respectively), cytochrome *c* oxidase subunit 6A1 (COX6A1) and coenzyme Q8B (COQ8B). According to Liu et al. [78], the level of metallothioneins is reduced in HCC, a phenomenon that is associated with a reduced ability to detoxify heavy metals, and restoring the level of these proteins hinders metastasis both in vitro and in vivo. This is due to the ability of MT1X and MT2A to scavenge free radicals and metals, in particular zinc and copper [78], and to inhibit the activity of NF-κB [79]. We showed that the C_60_ nanofilm upregulated both MT1X and MT2A. It also increased the level of protein IV of the mitochondrial respiratory chain complex involved in the oxidative phosphorylation pathway, namely COX6A1 [80]. Reduced COX6A1 expression in hepatocytes causes apoptosis and liver damage. The reason for the reduced level of COX6A1 in hepatocytes is its degradation by autophagy [81]. It follows that the C_60_ nanofilm inhibited intracellular ROS production as a result of the upregulation of COX6A1 and thus could protect HepG2 cells from type II programmed cell death [82]. CoQ8 has a similar effect, protecting healthy cells from apoptosis caused by ROS generated after the administration of chemotherapeutic agents [83]. CoQ levels are increased by various forms of oxidative stress, such as exercise and cold adaptation [83], which may be associated with healthy compensatory mechanisms. The proteomic analysis confirmed that C_60_ nanofilm inhibits autophagy and reduces inflammation and oxidative stress, and it consequently restores the epithelial (dormant) phenotype of HCC cells.

Research and therapies targeting EMT have certain limitations resulting from the complexity of EMT regulatory networks, the dependence of the process on the microenvironment (growth factors and stiffness), changes in regenerative processes and homeostasis, acquisition of drug resistance, acceleration of proliferation, tissue heterogeneity in vivo, different in vitro/in vivo responses, and responses dependent on primary tumour variability. For this reason, many EMT inhibitors have failed clinical trials, including a TGF-β inhibitor (galunisertib). Moreover, even promising animal studies do not guarantee the effectiveness of EMT-reversing agents, as was the case with soracatinib [84]. The next step of our future research will be to prove the promotion of MET by fullerene in chicken embryos.

## 5. Conclusions

We assumed that using a C_60_ nanofilm as an element of the ECM scaffold would limit the invasion of the HCC cell lines HepG2 and SNU-449 but would not affect the non-cancerous HFF2 cell line. In the first panel of experiments, we showed that the C_60_ nanofilm has a low roughness of 2.6 nm and is a good candidate to promote the MET of cancer cells because it reduced the expression of the key inducer of the invasive phenotype—TGF—and has an antioxidant nature. In the second panel of experiments, we induced the invasive, mesenchymal phenotype of HepG2 and SNU-449 cells by treating them with a growth factor cocktail. When grown on C_60_ nanofilm, the invasive cells showed reduced two- and three-dimensional invasion and a reduced expression of MMPs involved in degrading the ECM. In addition, we found that the C_60_ nanofilm is involved in the recovery mechanisms of the epithelial phenotype, that is, MET. The C_60_ nanofilm reduced cell invasion by changing the expression of the β-catenin → keratin → smad4 → snail/slug signalling pathway components. We also found that the recovery of the epithelial phenotype of the cells was associated with a reduction in autophagy and an increase in the enzymatic (catalase) and non-enzymatic (metallothionein) antioxidant systems. It is worth emphasising that fullerene exhibits anti-inflammatory properties by inhibiting elements of the NF-κB pathway. Based on our findings, we can undoubtedly conclude that C_60_ nanofilm induces EMT reversion, changing the invasive phenotype of liver cancer cells to a non-invasive phenotype. When introduced to tumour contact sites, fullerene may be a potential way to limit the malignancy or metastasis of cancer cells.

## Figures and Tables

**Figure 1 cancers-15-05553-f001:**
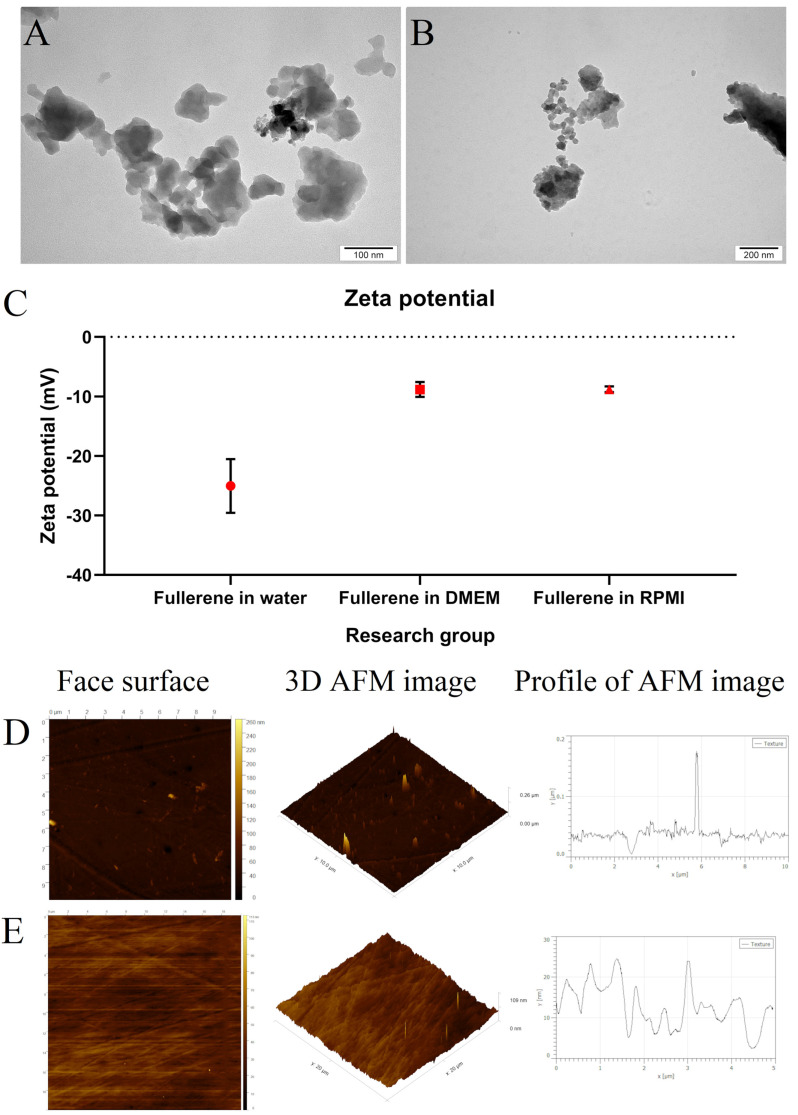
(**A**,**B**) Characterisation of fullerene nanoparticles and nanofilm using transmission electron microscopy ((**A**), scale bar: 100 nm; (**B**), scale bar: 200 nm), (**C**) a Zetasizer and (**D**,**E**) atomic force microscopy. Comparison of the surface roughness of (**D**) an uncoated polystyrene plate and (**E**) a fullerene-coated polystyrene plate.

**Figure 2 cancers-15-05553-f002:**
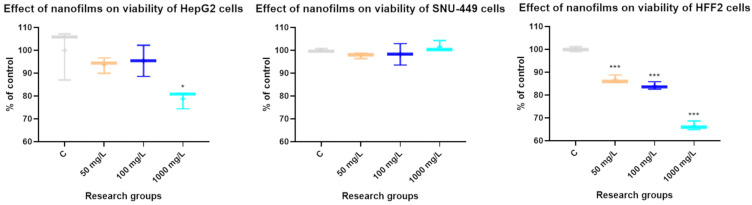
Mitochondrial dehydrogenase activity of HepG2, SNU-449, and HFF2 cells after growth on uncoated and C_60_-coated plates at a concentration of 50, 100, and 1000 mg/L using the MTT assay. Data are presented in the form of box-whisker charts. The medians are shown as a middle, horizontal line, and the percentiles (2.5th, 25th, 50th, 75th, and 97.5th) as vertical lines. The whiskers indicate the minimum and maximum values. Statistical significance is indicated by asterisks: * *p*  <  0.05 and *** *p*  <  0.001.

**Figure 3 cancers-15-05553-f003:**
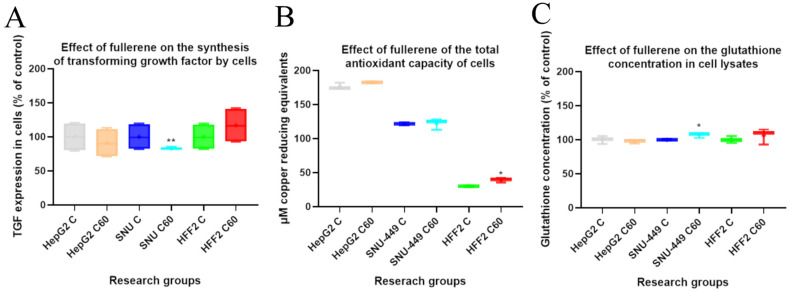
(**A**) The level of transforming growth factor β1 (TGF-β1), (**B**) the total antioxidant capacity and (**C**) the level of glutathione in the cell lysates after incubating cells for 24 h on C_60_ nanofilm. Data are presented in the form of box-whisker charts. The medians are shown as a middle, horizontal line, and the percentiles (2.5th, 25th, 50th, 75th, and 97.5th) as vertical boxes/lines. The whiskers indicate the minimum and maximum values. Statistical significance is indicated by asterisks: * *p*  <  0.05 and ** *p*  <  0.01.

**Figure 4 cancers-15-05553-f004:**
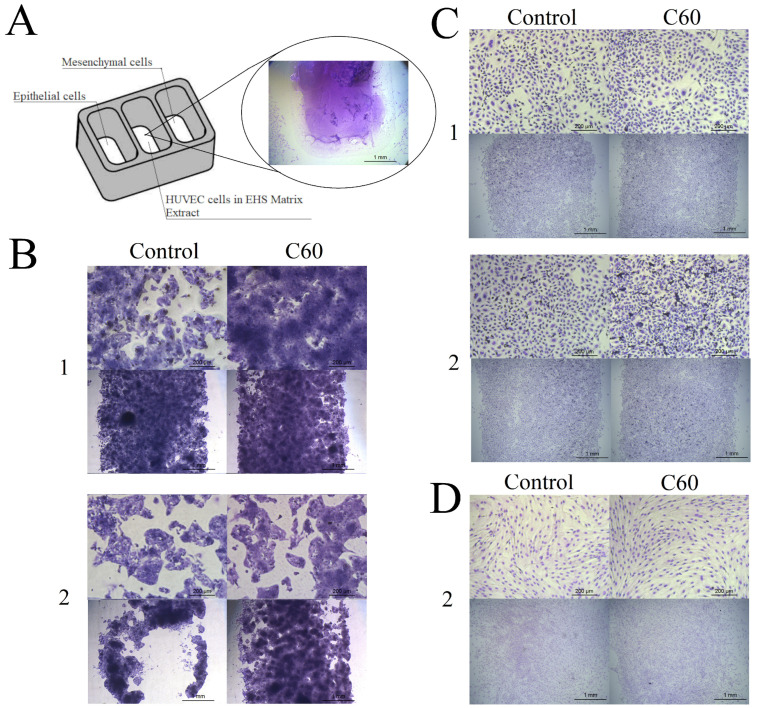
(**A**) Scheme of the two-dimensional invasion experiment. The central well contained a human umbilical vein endothelial cell, extracellular matrix proteins and C_60_ nanofilm. The micrographs show the morphology of (**B**) HepG2, (**C**) SNU-449 and (**D**) HFF2 cells with the (1) epithelial and (2) mesenchymal phenotypes cultured on an ordinary polystyrene plate (control) and a polystyrene plate coated with C_60_ nanofilm after incubation for 48 h (scale bar: 200 µm). In panels (**B**–**D**), the lower micrographs show cell growth towards the cell-free gap (scale bar: 1 mm).

**Figure 5 cancers-15-05553-f005:**
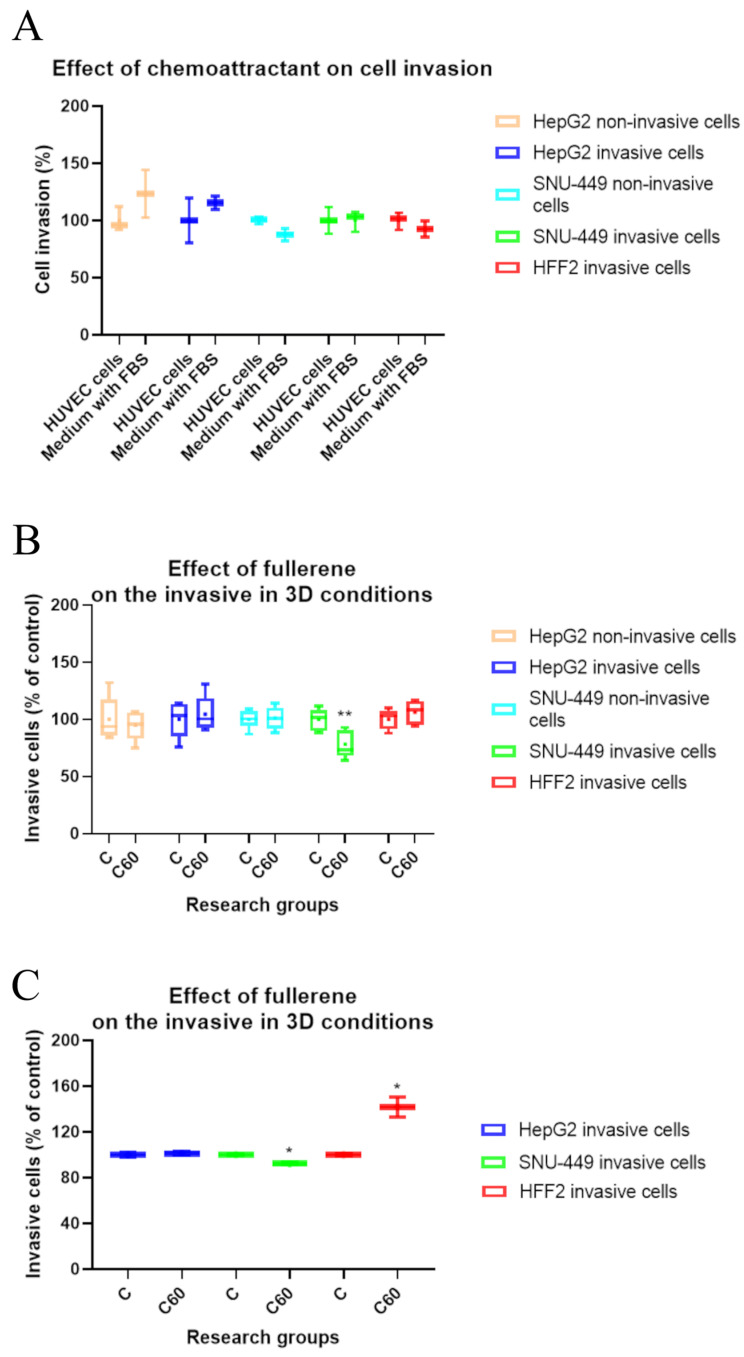
Cell invasion through a polycarbonate membrane (with 8 µm pores) coated with proteins of the Engelbreth–Holm–Swarm (EHS) mouse tumour basement membrane. (**A**) The effect of chemo-attractants on three-dimensional cell invasion after incubation for 24 h and staining with CyQuant GR Dye. Cell invasion is expressed relative to the two-cell-type co-culture assay. (**B**) The effect of C_60_ nanofilm on the invasive capacity of epithelial (non-invasive) and mesenchymal (invasive) cells after incubation for 24 h and staining with CyQuant GR Dye. (**C**) The effect of C_60_ nanofilm on invasion of mesenchymal cells after incubation for 72 h and lysis of cells stained with crystal violet. Data are presented in the form of box-whisker charts. The medians are shown as a middle, horizontal line, and the percentiles (2.5th, 25th, 50th, 75th, and 97.5th) as vertical boxes/lines. The whiskers indicate the minimum and maximum values. Statistical significance is indicated by asterisks: * *p*  <  0.05 and ** *p*  <  0.01.

**Figure 6 cancers-15-05553-f006:**
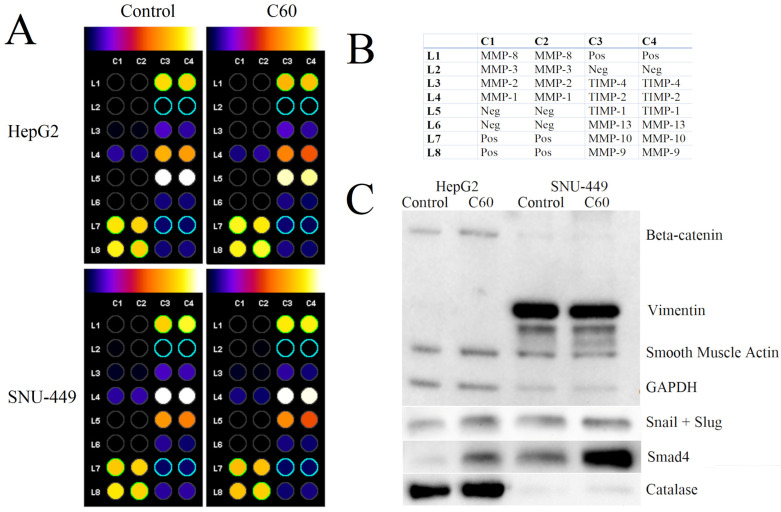
(**A**) The metalloproteinase (MMP) and tissue inhibitor of metalloproteinase (TIMP) profile in lysates from cells grown on uncoated polystyrene plates and plates coated with C_60_ nanofilm. Unmodified protein membranes are shown in Appendix A. (**B**) The table shows the location of individual MMPs on the protein membrane. The positive control (pos) was biotin-conjugated IgG. Protein expression was evaluated using densitometry software for a semiquantitative comparison using ImageJ 1.54d software. The background was removed with Protein Array Analyzer for the ImageJ 1.54d software. The colour scale shows protein levels from lowest (black) to highest (white). (**C**) Western blot analysis of beta-catenin, vimentin, smooth muscle actin, glyceraldehyde 3-phosphate dehydrogenase (GAPDH), snail/slug, Smad4 and catalase. GAPDH and smooth muscle actin were used as a loading control.

**Figure 7 cancers-15-05553-f007:**
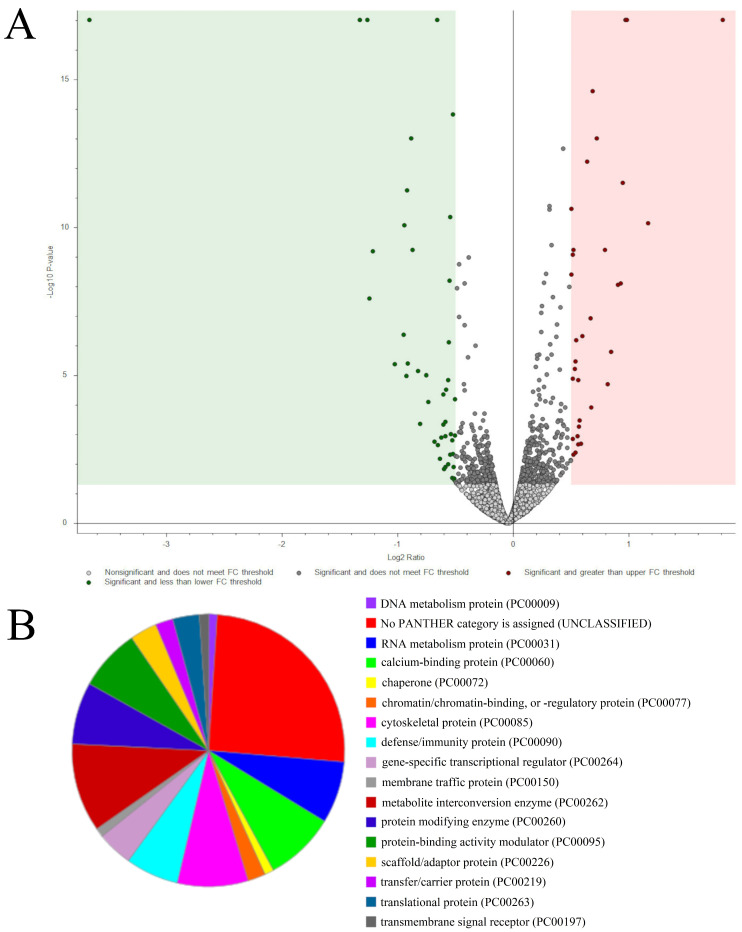
The effect of the growth of HepG2 cells on C_60_ nanofilm for 48 h on the change in expression of proteins based on mass spectrometry. (**A**) A volcano plot of 6295 identified proteins from HepG2 cells based on mass spectrometry. Significant changes in intracellular protein expression after culturing cells on C_60_ nanofilm compared with the control (fold change = 1.4, n = 3 per group, *p* ≤ 0.05) are indicated by green (downregulation) and pink (upregulation) colours. (**B**) The pie chart of protein classes that changed after growth on C_60_ nanofilm made with the Panther Classification System 18.0 software.

**Figure 8 cancers-15-05553-f008:**
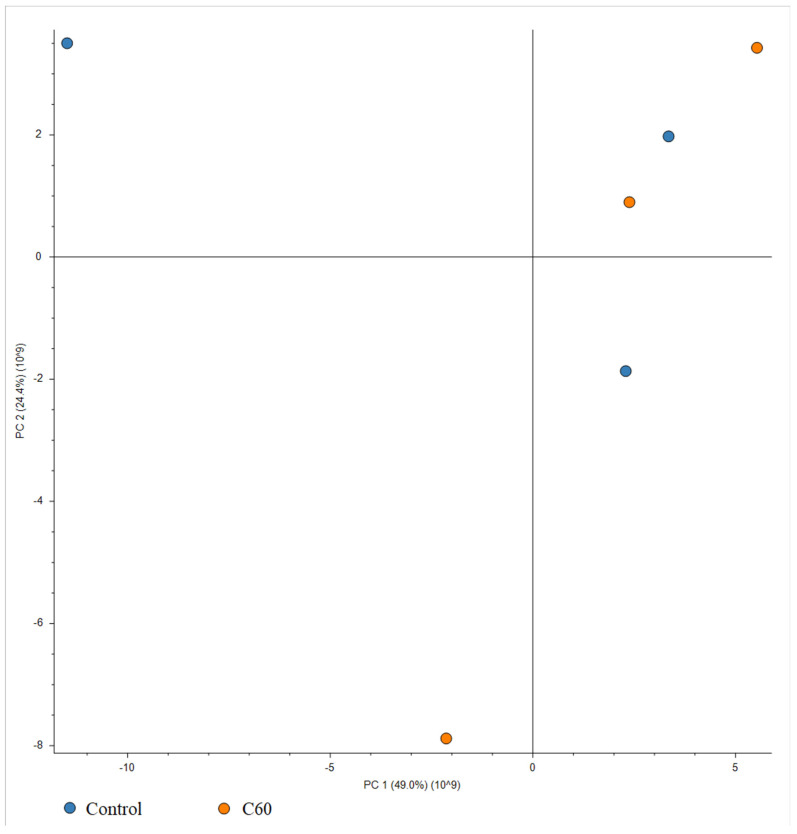
Principal component analysis of the general proteome of HepG2 cells. The score plot was prepared with all identified proteins. The blue circles indicate the control group (growth on uncoated plates), and the orange circles indicate the experimental group (growth on C_60_ nanofilm). Principal components 1 and 2 (PC1 and PC2) explain 49.0% and 24.4% of the sample variation, respectively.

**Figure 9 cancers-15-05553-f009:**
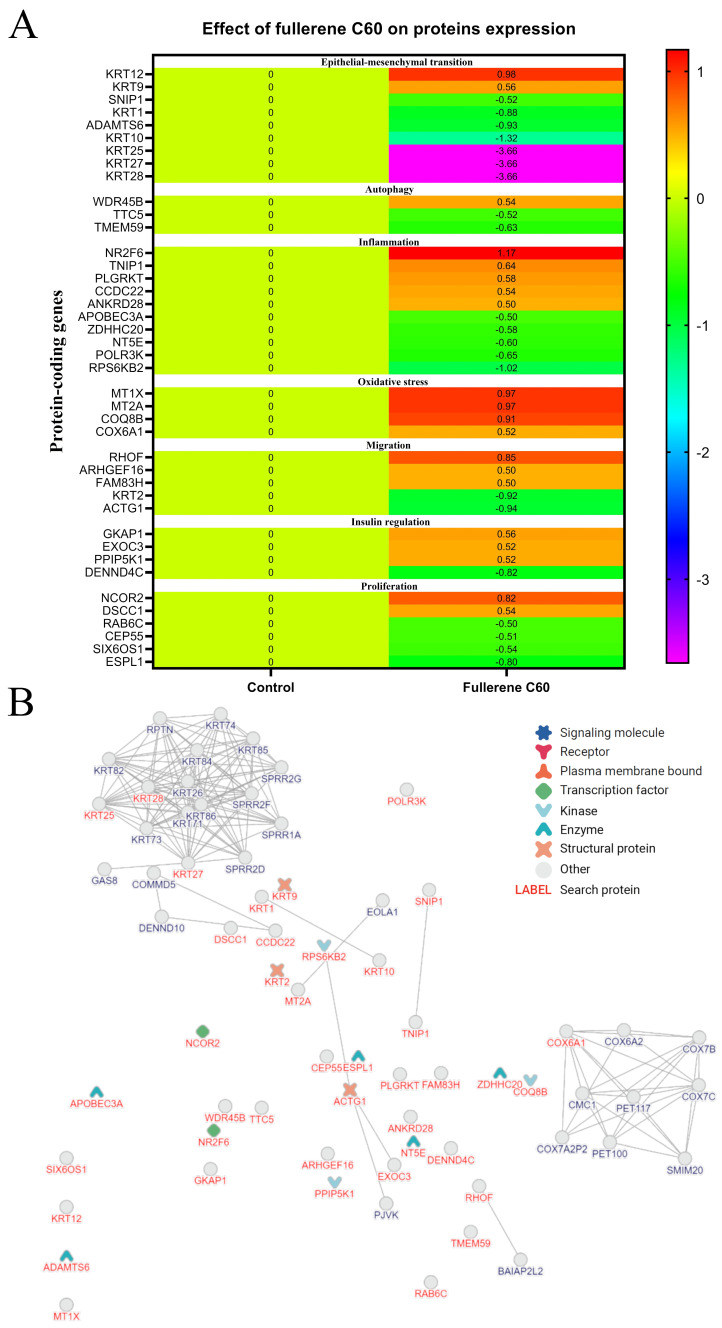
The effect of C_60_ nanofilm on the expression of 41 key proteins of HepG2 cells. (**A**) Heat map of the expression of 41 key proteins of HepG2 cells cultured for 48 h on an uncoated polystyrene plate (control) and on a C_60_ nanofilm-coated polystyrene plate. The proteins were identified, and their expression was quantified by mass spectrometry. Each column of the heat map represents the research group (control or fullerene C_60_), and each row represents the symbol of the gene that encodes the study protein. The results are presented as fold changes expressed as the log_2_ ratio (ratio: fullerene vs. control). (**B**) The protein–protein network in HepG2 cells. The key 41 proteins were selected from 96 proteins with altered expression after growth on C_60_ nanofilm. These 41 proteins are marked in red font and are divided into protein categories. In the network, neighbouring proteins are marked in black font. Abbreviations: KRT, keratin; SNIP1, smad nuclear-interacting protein 1; ADAMTS6, a disintegrin and metalloproteinase with thrombospondin motifs 6; WDR45B, WD repeat domain phosphoinositide-interacting protein 3; TTC5, tetratricopeptide repeat protein 5; TMEM59, transmembrane protein 59; NR2F6, nuclear receptor subfamily 2 group F member 6; MT2A, metallothionein-2; TNIP1, TNFAIP3-interacting protein 1; PLGRKT, plasminogen receptor KT; CCDC22, coiled-coil domain-containing protein 22; ANKRD28, serine/threonine-protein phosphatase 6 regulatory ankyrin repeat subunit A; APOBEC3A, DNA dC→dU-editing enzyme APOBEC-3A; ZDHHC20, palmitoyltransferase ZDHHC20; NT5E, 5’-nucleotidase; POLR3K, DNA-directed RNA polymerase III subunit RPC10; RPS6KB2, ribosomal protein S6 kinase beta-2; MT1X, metallothionein-1X; COQ8B, atypical kinase COQ8B; COX6A1, cytochrome c oxidase subunit 6A1; RHOF, rho-related GTP-binding protein RhoF; ARHGEF16, rho guanine nucleotide exchange factor 16; FAM83H, protein FAM83H; ACTG1, Actin, cytoplasmic 2; GKAP1, g kinase-anchoring protein 1; EXOC3, exocyst complex component 3; PPIP5K1, inositol hexakisphosphate and diphosphoinositol-pentakisphosphate kinase 1; DENND4C, DENN domain-containing protein 4C; NCOR2, nuclear receptor corepressor 2; DSCC1, sister chromatid cohesion protein DCC1; RAB6C, ras-related protein Rab-6C; CEP55, centrosomal protein of 55 kDa; SIX6OS1, protein SIX6OS1, ESPL1, separin.

**Table 1 cancers-15-05553-t001:** Information about the primary antibodies used for Western blotting.

Primary Antibody	Molecular Mass (kDa)	Catalogue Number	Company
Glyceraldehyde-3-phosphate dehydrogenase (GAPDH)	36	ab157392	Abcam
Smooth muscle actin	42		
β-catenin	92		
Vimentin	54		
Snail and Slug	29	ab180714	
Catalase	60	ab179843	
Smad4	60	PA5-34806	Thermo Fisher Scientific

## Data Availability

The data presented in this study are available on request from the corresponding author.

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
