# Peer review of "Influence of C60 Nanofilm on the Expression of Selected Markers of Mesenchymal–Epithelial Transition in Hepatocellular Carcinoma"

_cancers, 2023, doi:10.3390/cancers15235553_

Round 1

Reviewer 1 Report

Comments and Suggestions for Authors

Comments for paper

Comments:

(1)    Figures 2, 3 and 5 should be presented as box-whisker charts, showing mean values from independent biological experiments and not just bar charts.

(2)    The width of bar charts in Figures 5a and 5b should be enlarged for clarity.

 Line 382: The main cytokine involved in EMT induction is TGF-β1; thus, we investigated the 382 effect of fullerene nanofilm on the secretion of this factor by cells.

 Comments:

(3)    I agree with the authors that TGF-β1 is the main cytokine that drives EMT but to make the work more robust it will be good to check the expression of one or more cytokines that induces EMT for example the tumour necrosis factor-alpha (TNF-α) in the three cell lines.

(4)   A complete list of abbreviations used in the manuscript should be included.

Author Response

Dear Reviewer, 

Best regards,

The Authors

Reviewer 2 Report

Comments and Suggestions for Authors

In this work authors demonstrated that the use of fullerene C60 nanofilm can be used inhibit liver cancer cell invasion by restoring their non-aggressive, epithelial phenotype. To validate this approach authors performed extensive testing’s and shown the results to prove the hypothesis. This is an interesting work; I feel this work will be very helpful for the basic researchers and clinicians who are working in the Cancers. In addition, this is well written paper, but discussion is pretty long, may authors revise it and reduce the length and bit more succinate. I am happy to recommend for the publication.

Minor comments:

Figure 6 A and B need to link in such a way which panel in Figure6A is C1, C2 and so on.

Comments on the Quality of English Language

Discussion section need to be edited

Author Response

(The authors gave the same response as above.)

Reviewer 3 Report

Comments and Suggestions for Authors

In this research, the authors evaluated the influence of C60 Nanofilm on the expression of selected markers of mesenchymal-epithelial transition in hepatocellular carcinoma. Generally, it’s meaningful and interesting research. In my opinion, the current version of this manuscript fits the scope of the Cancers and could be accepted after minor revision.

My specific comments are in detail listed below:

1.     The quality of the figures listed in this paper is of low quality. A better clear version should be added.

2.     In the discussion part, what’s the drawback of such design? The authors should discuss the defects or limitation as well.

3.     In the introduction part, the merits of C60 Nanofilm compared with some other tumor metastasis inhibition nanosystem or drugs should be emphasized and more clearly discussed. Some references could be added to this part.

4.     If possible, flow cytometry assay may could better evaluate the tumor cell killing capacity of this nanosystem.

5.     Was the tumor cell growth affected by this nano-design? If possible, the authors could discuss or prove it. Some references could be added to this part including 10.1016/j.jconrel.2022.11.004.

6.     How was the tumor distribution behavior of this nanosystem? The in vivo bio-distribution behavior could be evaluated by the small animal live imaging system.

7.     Some minor mistakes exist in this paper. The authors should carefully polish it.

Author Response

(The authors gave the same response as above.)
